# Effects of different sponge implantation methods of negative pressure wound therapy on wound healing of deep surgical site infection after spinal surgery

**Jingming Wang, Hao Xing, Zhengqi Chang**◉*

Department of Orthopedics, 960th Hospital of PLA, Jinan, China

* 26766771@qq.com

**Data Availability Statement:** All relevant data are within the paper and its Supporting Information files.

## Abstract

### Purpose

After spinal surgery, negative pressure wound treatment (NPWT) improves deep surgical site infection (DSSI) wound healing. This research compared the healing benefits of two sponge implantation strategies in NPWT for DSSI.

### Methods

21 patients with DSSI utilized NPWT to improve wound healing following spine surgery were followed from January 1, 2012 to December 31, 2021. After antibiotic treatment failure, all these patients with DSSI received extensive debridement and NPWT. They are grouped by sponge placement method: centripetal reduction and segment reduction. The two groups' hospital stays, NPWT replacement frequency, wound healing time, healing speed, and quality of wound healing (POSAS score) were compared.

### Results

All patients had been cured by the end of December 2022, and the mean follow-up time was 57.48 ± 29.6 months. Surgical incision length did not vary across groups (15.75±7.61 vs. 15.46±7.38 cm, P = 0.747). The segmental reduction approach had shorter hospital stay and NPWT treatment times than the centripetal reduction method (39.25±16.04 vs. 77.38 ±37.24 days, P = 0.027). Although there is no statistically significant difference, the mean wound healing duration of segmental reduction group is faster than that of centripetal reduction group (0.82±0.39 vs 0.45±0.28 cm/d, P = 0.238), wound healing quality (POSAS) (33.54±8.63 vs 48.13±12.17, P = 0.408) is better in segmental reduction group, and NPWT replacement frequency (2.62 ± 1.04 vs 3.88 ± 1.25, P < .915) is smaller in segmental reduction group.

**Funding:** The author(s) received no specific funding for this work.

**Competing interests:** The authors have declared that no competing interests exist.

## Conclusions

NPWT heals wounds and controls infection. Segmental reduction method accelerates wound healing, reduces hospital stay, and improves wound quality compared to central reduction method.

## Introduction

Surgical site infection (SSI) is defined as a surgery-related infection that occurs within 30 days of surgery without implants and within 1 year of surgery with implants [1]. Pedicle screw fixation is currently the classic surgical approach for the treatment of spinal diseases, including spinal degenerative diseases, spinal deformities, and spinal tumors. Incision infection after spinal surgery is the most common complication of spinal internal fixation, with an incidence of 2–4.15%. SSI significantly prolongs the length of hospital stay, increases the cost of treatment, and reduces patient satisfaction [2, 3].

Most superficial SSIs can be well controlled with conservative treatment, while deep infections often require surgery. Negative pressure wound therapy has been proved to be an effective method to prevent and control infection [4–6]. Although the infection can be controlled effectively, the infection cause large defects in the incision, and it takes a long time to heal in the later stage, which significantly prolongs the treatment time. Therefore, how to quickly promote incision healing and improve the quality of incision healing is still an urgent clinical problem to be solved, no standard protocol has been proposed to guide how to place the sponge in NPWT.

Since 2012, we have used the NPWT to treat deep infection after spinal surgery. All patients did not undergo internal fixation removal, and achieved good results [7]. After infection control, we used the NPWT technique to promote wound healing with satisfactory results. In this study, we used two different sponge placement methods, the centripetal reduction method and the segmental reduction method. The healing effect of NPWT on infected wounds after spinal surgery and the two placement methods of NPWT sponge were compared and analyzed to propose a better sponge placement method.

## Methods

### Patient population

We retrospectively analyzed 23 patients admitted to the Department of Orthopedics in our hospital from January 1, 2012 to December 31, 2021 who were treated with NPWT for deep spine surgical infection. All patients had incision nonunion on admission, accompanied by incision pain and pus or viscous secretions, and were diagnosed as deep postoperative spinal infection by imaging examination or bacterial culture. All patients were treated with debridement + NPWT after antibiotic treatment failed. After infection control, NPWT was continued to promote wound healing. The incisions healed well and the patients were monitored at the outpatient department after leaving the hospital. Excluding 2 patients lost to follow-up, a total of 21 patients were included in this study, including 11 males and 10 females, with an average age of 55.83±17.41 years, and all patients had been cured by the end of December 2022. This study has been approved by the 960[th] hospital of PLA.

## Antibiotic treatment

All patients were treated with empiric antibiotics before admission to our hospital, and were admitted to our hospital for treatment after ineffective treatment. After admission, the secretions of all patients were collected for bacterial culture, and antibiotics were used to fight infection according to experience. After bacterial culture and drug susceptibility results feedback, sensitive antibiotics were used for anti-infection treatment. After the patient's wound was healed, oral antibiotics were continued outside the hospital for 6 weeks.

## Surgical technique

All patients underwent SSI debridement under general anesthesia, that is, after thorough debridement, the NPWT sponge was placed to fully cover the internal fixation, and the sponge was placed into the intervertebral space. The sponge is inserted into the intervertebral space from the outside of the titanium rod and sutured to the main sponge to prevent detachment. The main sponge is placed on both sides of the spinous process like a dam. If the dura is exposed, trim the exposed area into an arch. If the dura is covered with granulation tissue, the primary sponge can be placed on the dura and nerve roots [7]. Continue to maintain negative pressure drainage after surgery and renew the sponge every 5–7 days under local anesthesia.

CRP decreased to 1/2 before admission, and the patient's fever symptoms disappeared, which was regarded as effective infection control. After infection control, infection treatment enters the wound healing phase. Two different sponge placement method were used at this stage. The first method, called the centripetal reduction method, involves taking the entire sponge and placing it at the incision. After the wound surface was gradually reduced, it was replaced with a smaller sponge, which was reduced from the side without fenestration to the side with clear fenestration, and the reduction did not exceed 3 cm each time (6 cm in total on both sides). The second method is called the segmented reduction method, that is, the sponge is trimmed into several pieces, each sponge is 1cm wide and 1cm thick, and placed at intervals of 2cm to the incision. Afterwards, the sponge is removed according to the wound healing (**Figs 1–3**).

## Data collection

The general condition and postoperative recovery indicators of the patients were collected. Basic patient information (eg, gender, age, BMI, comorbidities), surgery-related conditions (eg, number of surgical segments, wound length), as well as bacterial culture, length of hospital stay, and prognostic factors (VAS score, patient satisfaction, infection recurrence) were recorded. At the same time, wound healing-related parameters including wound healing speed and wound healing quality (Patient-Observer Scar Assessment Scale, POSAS) were recorded [8]. The patients were followed up for at least 1 year after discharge, and the relevant indicators were counted to assess the condition.

## Statistical analysis

SPSS 22.0 statistical software was used for analysis. Enumeration data were expressed as frequency, measurement data were expressed as mean, paired $t$-test or $\chi^2$ test was used for comparison between groups, and $P < 0.05$ was regarded as statistically significant.

## Results

A total of 21 patients with deep incision infection after spinal surgery were included in this study, including 11 males and 10 females, with an average age of 55.83 ± 17.41 years. No

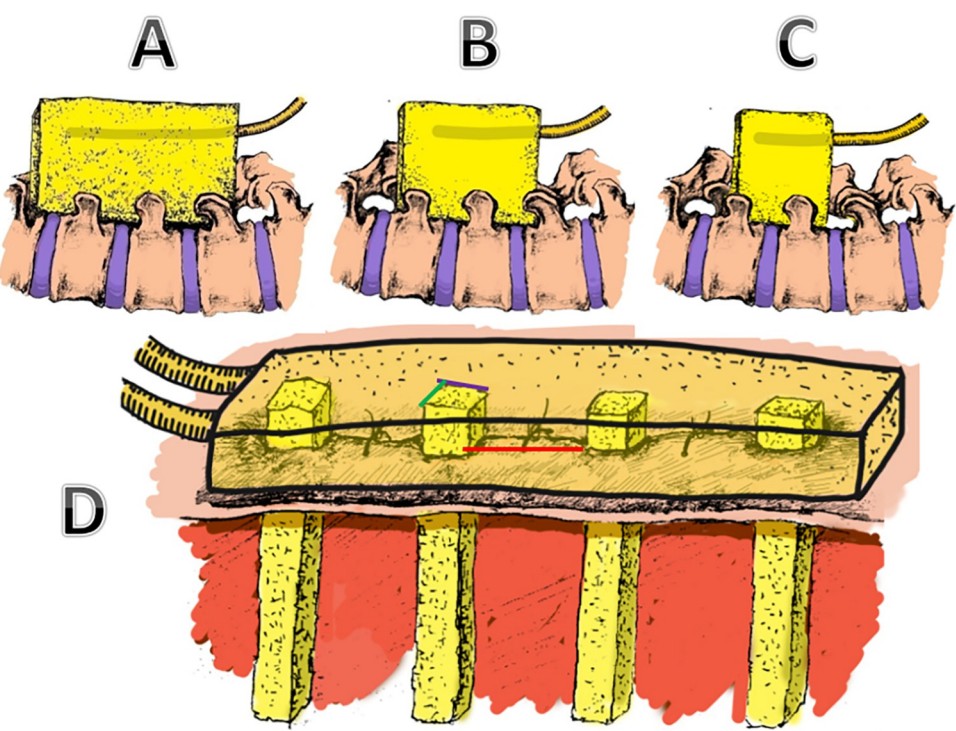

**Fig 1. Schematic diagram of two NPWT sponge placement methods.** (A-C) Centripetal reduction method. Take the whole sponge and place it at the incision. After the wound surface is gradually reduced, replace it with a smaller sponge. (D) Subsection reduction method. Trim the sponge into several pieces, each with a width of 1cm and a thickness of 1cm. They were placed at intervals to the incisions, sutured at 2 cm intervals, and the separated sponges were connected to the main sponge.

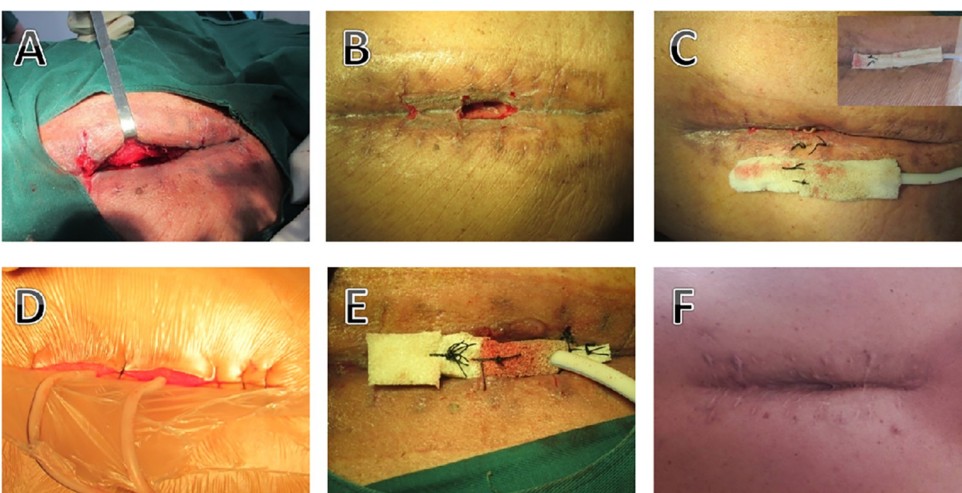

**Fig 2. A typical case of the centripetal reduction method.** The patient was a 59-year-old male who underwent L5-S1 spinal decompression and internal fixation in another hospital. One month after the operation, pus from the incision was seen in our hospital. The bacterial culture was Pseudomonas aeruginosa. (A-C) Thorough debridement of the incision and placement of a sponge to cover the wound. (D) Connect the negative pressure drainage device for continuous negative pressure drainage. (E) At intervals of 5–7 days, the sponge was replaced, and the size of the sponge was gradually reduced according to the wound healing. (F) Wound after healing.

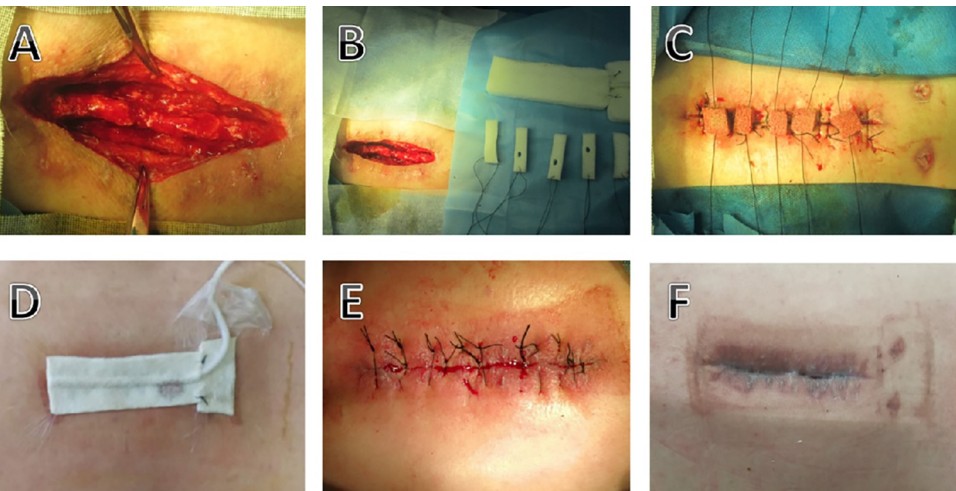

**Fig 3. Typical case of segment reduction method.** A 43-year-old man underwent lumbar spondylolisthesis reduction and internal fixation for lumbar spondylolisthesis. Six months after the operation, he developed an incision sinus with pus and was referred to our hospital. (A) The incision was thoroughly debridement, and the deep wound was covered with sponge. (B, C) The sponge was trimmed into several pieces, each with a width of 1 cm and a thickness of 1 cm, and placed at intervals to the incision. Sutures were made at 2 cm intervals, and the separate sponges were attached to the main sponge. (D) Connect the negative pressure drainage device for continuous negative pressure drainage. (E) At intervals of 5–7 days, the sponge is replaced. When the deep tissue is well covered, the incision is sutured as appropriate. (F) The wound scar healed well.

internal fixation was removed in all patients, and the wounds healed well and all patients were discharged from hospital. The general information of the patients is shown in Table 1.

There was no significant difference in the BMI, surgery levels and length of surgical incision between the two groups (P>0.05). The segmental reduction method showed less hospital duration (39.25±16.04 vs. 77.38±37.24 days, $P = 0.027$) and NPWT treatment time (20.62 ± 8.41 vs. 39.25 ± 16.04 days vs. 39.25 ± 16.04 days, $P = 0.027$) than the centripetal reduction method, the hospital stay of patients in centripetal reduction group is nearly double of the patients in segmental reduction group. Although there is no significant difference in the average wound healing speed (0.82±0.39 vs 0.45±0.28 cm/d, P = 0.238), NPWT replacement frequency (2.62 ± 1.04 vs 3.88 ± 1.25, P = 0.915) and wound healing quality (POSAS) (33.54±8.63 vs 48.13±12.17, P = 0.408), the wound found to be found to be quick and of high quality in segmental reduction group. (Table 2).

## Discussion

The incidence of SSI after spinal surgery is 1–5%, and the incidence of deep infection is 0.81% [9, 10]. Once postoperative deep infection occurs, the patient's treatment cycle is significantly increased, and even death [2, 3, 10, 11]. Studies have shown that the risk factors of incision

**Table 1. Baseline characteristics of participants.**

|  | Male/Female | Age | BMI | Laboratory examination | | | Comorbidities | Surgery Levels | Length of incision (cm) |
|---|---|---|---|---|---|---|---|---|---|
|  |  |  |  | WBC($\times 10^9$/L) | ESR (mm/h) | CRP (mg/L) |  |  |  |
| 21 | 11/10 | 55.83±17.41 | 24.54±3.15 | 9.43±4.04 | 61.42±37.23 | 41.64±50.6 | 9/21 | 3.24±1.64 | 15.57±7.28 |

Comorbidities: 4 patients had diabetes, 2 had hypertension, 1 had rheumatoid arthritis (hormone use history), 1 had hepatitis B, and 1 had atrial fibrillation and old cerebral infarction.

**Table 2. Clinical results between two groups.**

| Group | centripetal reduction (n = 8) | segmental reduction (n = 13) | P |
|---|---|---|---|
| **Preoperative data before NPWT** | | | |
| **Age** | 56.86±21.45 | 57.15±10.02 | 0.046* |
| **BMI** | 26.2±2.83 | 23.54±10.00 | 0.806 |
| **Surgery segments** | 3.13±1.36 | 3.42±1.88 | 0.796 |
| **Length of incision (cm)** | 15.75±7.61 | 15.46±7.38 | 0.747 |
| **Clinical results** | | | |
| **NPWT replacement frequency** | 3.88±1.25 | 2.62±1.04 | 0.915 |
| **NPWT treatment duration (day)** | 39.25±16.04 | 20.62±8.41 | 0.015* |
| **Hospital duration(day)** | 77.38±37.24 | 34.92±19.01 | 0.027* |
| **Wound healing speed (cm/day)** | 0.45±0.28 | 0.82±0.39 | 0.238 |
| **Quality of healing (POSAS)** | 48.13±12.17 | 33.54±8.63 | 0.408 |

The wound healing speed was after infection control, and after entering the wound healing stage, the incision length/healing time was counted; the wound healing quality was evaluated by POSAS (Patient and Observer Scar Assessment Scale), with a score of 6–110, and 6 was a normal incision.

infection associated with patients after spinal surgery are mainly obesity, diabetes, and tumors, while the risk factors associated with surgery are mainly the posterior approach, more surgical segments, internal fixation, and more bleeding [12–14]. In some SSI patients, the internal fixation needs to be removed during the treatment process, resulting in the loss of spinal stability and the failure of the operation.

Perioperative antibiotics, adequate incision drainage, and regular dressing changes after surgery are important factors to reduce postoperative infection [15, 16]. For superficial infections, the application of sensitive antibiotics is able to achieve satisfactory clinical efficacy. However, for deep infection, the effect of antibiotics alone to fight infection is limited. When conservative treatment fails, surgical debridement is required. The extensive debridement was under general anesthesia, due to the poor condition of SSI patients, hypotension might occur during surgery, which is associated with increased postoperative morbidity and mortality, intraoperative monitoring, early warning system, inotropes to increase oxygen delivery and hemodynamic algorithm can be used to reduce intraoperative hypotension and oxidative stress [17, 18]. After debridement surgery, commonly used treatment methods include incision debridement and suture plus long-term dressing change, incision irrigation and drainage, and NPWT. Debridement and suture plus long-term dressing change always lead to a significant increase in the treatment cycle. And when the tissue defect is massive, a satisfactory therapeutic effect is often not achieved. Incision irrigation and drainage have a certain therapeutic effect on the control of deep infection, but cannot improve postoperative tissue defects.

NPWT has been widely used in the treatment of acute, subacute, and chronic infections of extremity wounds. In recent years, NPWT technology has been gradually applied in the treatment of infection after spinal surgery, and satisfactory therapeutic effects have been achieved. Canavese reported 17 cases of SSI after scoliosis surgery using NPWT. After treatment with NPWT, all 17 patients had satisfactory wound healing and no internal fixation removal, and finally achieved satisfactory clinical results and high-quality spinal fusion [19]. Ousey analyzed 10 retrospective studies and 4 case reports, suggested that NPWT is able to achieve satisfactory results in SSI after spinal surgery. At the same time, it should be noted that the contraindications to the application of NPWT are: active cerebrospinal fluid leakage or active bleeding, a postoperative spinal incision in patients with malignant tumors, and patients who are allergic to NPWT sponge [20].

In this study, patients with deep surgical infection after spinal surgery were treated with NPWT, satisfactory results were obtained, and all patients had high-quality wound healing. Compared with the traditional operation group, the NPWT group significantly improved the positive rate of bacterial culture, effectively avoided the removal of internal fixation, shortened the hospitalization time, reduced the hospitalization cost, and reduced the recurrence rate. As the protocol of sponge placement has not been proposed, we proposed two different methods: centripetal reduction and segmental reduction. Comparing the two groups, the segmental reduction method demonstrated faster wound healing speed, shorter treatment and hospitalization time than the central reduction method. Accelerating the wound healing speed can better reduce the hospitalization time, reduce hospitalization costs, and improve patient satisfaction. Although infection control and tissue repair co-exist during treatment, the emphasis varies at different stages. In the course of treatment, according to the development of the patient's condition, we artificially divide the treatment process into two stages: the infection control stage and the tissue defect repair stage. The objective of the phase of infection control is to restrict the infection. Postoperative pain in patients treated with NPWT could be significantly improved on the 3rd postoperative day [7]. Continuous NPWT may guarantee proper drainage, and the acquired drainage fluid could be cultured multiple times, which increases the positive culture rate and directs the administration of sensitive antibiotics efficiently. In this period, attention should be paid if the patient complained of headache. As active cerebrospinal fluid leakage is one of contraindications of NPWT, NPWT was not used if there was CSF leakage. A thorough history, exam and diagnostic workup are necessary to distinguish secondary headaches from preexisting conditions to accurately diagnose, treat and prevent complications [21].

When the CRP decreased by half or <30 mmol/L, the infection was considered to be effectively controlled. It has been reported in the literature that with NPWT treatment, the infection was able to be effectively controlled within 12–13 days after surgery [22]. Our results are consistent with this result. The objective of the tissue repair phase at an infection site is to enhance wound healing. During the wound healing stage, we continued to use NPWT to promote tissue healing. All wounds healed well and the patient was discharged smoothly.

The possible factors that NPWT significantly improve wound healing are as follows. First, constant NPWT may enhance the local microenvironment, enhance the local blood supply, and promote wound healing. Second, the NPWT device may swiftly remove purulent effusion, necrotic tissue, and wound exudate, minimize tissue swelling, and lower local concentrations of inflammatory agents and proteases that inhibit wound healing [23, 24]. Finally, NPWT may accelerate the local blood supply, provide oxygen and nutrients required for normal wound healing, and promote wound healing. Morykwas proposed that the blood flow of the wound increases under negative pressure, and the peak value can reach 4 times of the baseline blood flow. Previous research has shown that NPWT may be considerably superior to conventional therapies in terms of wound and infection healing duration [24]. It should be highlighted that since the incision often has substantial tissue defects after debridement and the healing duration is more than one-month, frequent dressing changes will cause the patient discomfort and increase the medical staff burden. After a single application, the NPWT device may be stored for five to seven days without the requirement for redressing. It is possible to replace NPWT at the bedside. The surgery is quite straightforward, which increases patient satisfaction and decreases medical staff burden.

The negative pressure value is usually set to -125 mmHg in the application of spinal surgery incision [25]. For patients with pain during drainage, the negative pressure value can be reduced to -40 to -80 mmHg until the pain is relieved [20]. For patients with risk factors such as advanced age, malnutrition, or receiving anticoagulation therapy, the initial negative

pressure value can be set at -75 to -100 mmHg, and then gradually increased to -125 mmHg according to the patient's tolerance [26].

Regarding wound healing and sponge placement, there is no single standard at present. To prevent dead space, the treatment principle is to cover the wound with a sponge. In our clinical work, we applied two methods of placing sponges and conducted a comparative study. The initial operation of all patients was a posterior median incision. During debridement, in order to ensure complete debridement, the debridement range should be at least 5 mm or more of the lesion, and the postoperative wound surface was slightly longer than the original surgical incision. There are no technical obstacles between the two placing procedures since both can enable adequate drainage of the incision and prevent fluid accumulation. Compared with the traditional suture dressing group, continuous negative pressure drainage significantly reduces the wound healing time. Comparing the two groups, the segmental reduction method demonstrated faster wound healing than the central reduction method, and shorter treatment and hospitalization times than NPWT. After infection control, we suggested that segmental reduction of sponge placement should be used to accelerate the wound healing.

In addition, compared to the quality of wound healing, after the incision healed successfully using the centripetal reduction approach, local depressions occurred owing to inadequate tissue filling, the wound scar thickness was reduced, and tension and elasticity were poor. Especially for wounds following multi-level spinal fusion, this phenomenon is more noticeable. In patients who had segmental reduction, the thickness of the incision scar after wound healing is comparable to the thickness of the surrounding tissue, and the tension and flexibility of the scar are enhanced. In individuals with longer scars, however, the quality of scar healing diminished.

Regarding the rate and quality of wound healing, there was no discernible difference between the two groups. Our study is primarily motivated by the tiny sample size. There was no statistically significant difference between the average healing rates of the segmental reduction method (0.82 ± 0.39 cm/day) and the centripetal reduction method (0.45 ± 0.28 cm/day). Patients and physicians rated the Patient-Observer Scar Assessment Scale (POSAS), with scores ranging from 6 to 110. A comparison was made between the incision skin and the surrounding skin. The higher the score, the worse the surface scar healing quality. Notably, despite the fact that all wounds were covered with new tissue following NPWT therapy, POSAS scores were consistently high. This demonstrates that even after effective treatment of a severe spinal infection, the patient's personal life will be affected by the persistent scar. In in addition to developing an effective approach for treating deep spinal incision infection, infection prevention remains an essential therapeutic topic.

There are also evident flaws in this research. First, this research is retrospective, no randomization was undertaken, and there is the possibility of bias. On the other hand, the number of included cases is small, therefore it is important to expand the number of cases in the follow-up research and to confirm the difference between the two sponge implantation procedures in a multicenter study. Third, the mechanism of two different sponge placement is not unknown, which needs to be studied further.

## Conclusion

NPWT is a feasible therapy for deep incision infection following spine surgery that reduces hospitalization time, and expenditures. The segmented reduction method, compared to the central reduction method, may expedite wound healing, minimize hospitalization duration, and enhance wound quality after infection management.

## Supporting information

**S1 Checklist. STROBE statement—checklist of items that should be included in reports of observational studies.**
(DOCX)

**S1 Dataset.**
(XLSX)

## Author Contributions

**Conceptualization:** Zhengqi Chang.

**Data curation:** Jingming Wang, Hao Xing.

**Investigation:** Jingming Wang, Hao Xing.

**Methodology:** Zhengqi Chang.

**Writing – original draft:** Jingming Wang.

**Writing – review & editing:** Zhengqi Chang.

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
