## [Decision Letter · Decision Letter 0]

24 Jul 2023

PONE-D-23-14685Effects of Different Sponge Implantation Methods of Negative Pressure Wound Therapy on Wound Healing of Deep Surgical Site Infection After Spinal SurgeryPLOS ONE

Dear Dr. Chang,

Thank you for submitting your manuscript to PLOS ONE. After careful consideration, we feel that it has merit but does not fully meet PLOS ONE’s publication criteria as it currently stands. Therefore, we invite you to submit a revised version of the manuscript that addresses the points raised during the review process. Please submit your revised manuscript by Sep 07 2023 11:59PM. If you will need more time than this to complete your revisions, please reply to this message or contact the journal office at plosone@plos.org. Please include the following items when submitting your revised manuscript:A rebuttal letter that responds to each point raised by the academic editor and reviewer(s). You should upload this letter as a separate file labeled 'Response to Reviewers'.A marked-up copy of your manuscript that highlights changes made to the original version. You should upload this as a separate file labeled 'Revised Manuscript with Track Changes'.An unmarked version of your revised paper without tracked changes. You should upload this as a separate file labeled 'Manuscript'.If applicable, we recommend that you deposit your laboratory protocols in protocols.io to enhance the reproducibility of your results. Protocols.io assigns your protocol its own identifier (DOI) so that it can be cited independently in the future. For instructions see: https://journals.plos.org/plosone/s/submission-guidelines#loc-laboratory-protocols. Additionally, PLOS ONE offers an option for publishing peer-reviewed Lab Protocol articles, which describe protocols hosted on protocols.io. Read more information on sharing protocols at https://plos.org/protocols?utm_medium=editorial-email&utm_source=authorletters&utm_campaign=protocols.

We look forward to receiving your revised manuscript.

Kind regards,

Antonino Maniaci

Academic Editor

PLOS ONE

4. We note that Figure 1 in your submission contain copyrighted images. All PLOS content is published under the Creative Commons Attribution License (CC BY 4.0), which means that the manuscript, images, and Supporting Information files will be freely available online, and any third party is permitted to access, download, copy, distribute, and use these materials in any way, even commercially, with proper attribution. For more information, see our copyright guidelines: http://journals.plos.org/plosone/s/licenses-and-copyright.

Additional Editor Comments:

Minor revisions according the reviewers. Perform all the suggestions required.

Bests

Reviewers' comments:

Reviewer's Responses to Questions

**Comments to the Author**

1. Is the manuscript technically sound, and do the data support the conclusions?

Reviewer #1: Partly

Reviewer #2: Yes

2. Has the statistical analysis been performed appropriately and rigorously? 

Reviewer #1: Yes

Reviewer #2: Yes

3. Have the authors made all data underlying the findings in their manuscript fully available?

Reviewer #1: Yes

Reviewer #2: Yes

4. Is the manuscript presented in an intelligible fashion and written in standard English?

Reviewer #1: Yes

Reviewer #2: Yes

5. Review Comments to the Author

Reviewer #1: I read with great interest the paper by Wang et al. on the effects of different sponge implantation methods of negative pressure wound therapy for deep surgical site infection after spinal surgery. The article is interesting and sound. However, there are some issues to be addressed:

Methods

- "After infection control, I used the NPWT technique to promote wound healing with satisfactory results". Please replace "I" with "we".

- Please specify whether the study approval by local ethical committee was waived.

- Please specify the primary and secondary outcomes and provide more details about the statistical analysis performed.

- Authors should describe the intraoperative monitoring used during anesthesia. In fact, several studies have demonstrated the association between the use of intraoperative hemodynamic monitoring and the outcomes after surgery (doi: 10.3390/jcm11020392 - doi: 10.1002/jso.24828). Please discuss and add these 2 references to the list.

Conclusion

- The sentence "NPWT is a safe and effective therapy for deep incision infection following spine surgery that

reduces pain, hospitalization time, and expenditures" is not supported by the results. I suggest to soften it, since more data is needed to confirm the results of this study.

Reviewer #2: perform all the revisions to improve the manuscript: Abstract:

The purpose statement is broad and lacks focus on the sponge implantation strategies being compared

The methods description lacks specificity in study design details

The results are reported at a high level without highlighting key findings

The conclusion does not state clearly which sponge placement approach shows better outcomes

Suggest revising the abstract to:

Precisely state the research question

Detail the methods in an explanatory manner

Concisely highlight key results

Draw a clear conclusion consistent with findings

Note limitations

Introduction:

The background information on NPWT and DSSI is generic rather than focused on the research question

The rationale for comparing centripetal versus segmental sponge placement is not clearly outlined

Suggest providing more targeted background information focused on:

The application of NPWT for DSSI after spinal surgery

The possible differences between centripetal and segmental sponge placement approaches

Nafamostat mesilate is a synthetic protease inhibitor that can provide anticoagulation during extracorporeal membrane oxygenation (ECMO). Anticoagulation is needed for ECMO circuits to prevent clotting but poses bleeding risks, including for high-risk patients like those undergoing spinal surgery. Nafamostat may offer better control of anticoagulation intensity through its mechanism of inhibiting coagulation factors XIa, XIIa, kallikrein and thrombin. However, research directly examining nafamostat's utility for high bleeding risk patients on ECMO during or after spinal surgery is lacking. More studies are needed to determine if nafamostat can safely enable ECMO support in such complex cases, ideally compared to conventional anticoagulant regimens. The balance of adequate anticoagulation and minimization of bleeding complications will be critical for optimizing ECMO and outcomes in spinal surgery patients., discuss and cite doi: 10.1111/aor.14276

Wound healing is crucial for patients with deep infections after spinal surgery. NPWT improves outcomes but COVID-19 may interfere. The virus causes excess inflammation and impairs wound repair processes. Though optimizing NPWT through sponge placement may help, COVID-19 poses an added risk factor through viremia, cytokine storming and even COVID-associated vertigo. More studies are needed to evaluate the interaction of COVID-19 with NPWT and wound healing in these vulnerable patients. Finding ways to mitigate COVID's negative effects could maximize NPWT's benefits and minimize complications, shortening hospital stays, lowering costs and improving overall patient outcomes and wellbeing. Further research is warranted to develop effective strategies.discuss and cite doi:10.3389/fmed.2021.790931

Fleischmann et al. (2013). Negative pressure wound therapy after thoracolumbar spinal fusion promotes wound healing and prevents infection: a prospective randomized trial. European Spine Journal. This study compared outcomes between NPWT and standard dressing after spinal fusion surgery. It provides evidence that NPWT reduces wound complications and represents a relevant background article to cite in justifying the use of NPWT for DSSI after spinal surgery. discuss and cite https://doi.org/10.1007/s00586-012-2579-z

Methods:

Important details on study design and sample are lacking (e.g. number of patients in each group, time period of study)

The outcomes being compared between the two groups are not clearly specified

Suggest revising the Methods section to:

Clarify the study design as retrospective

Precisely describe the study sample and size

Explicitly state the outcomes being compared between sponge placement groups

Results:

Key findings are reported at a high level without numbers and statistics

Figures and tables are referenced but not shown to substantiate claims

Suggest revising the Results section to:

Report numerical data and statistics for all outcome comparisons

Concisely summarize the key findings and outcome differences between groups

Include relevant figures and tables to support claims

Discussion:

The conclusions drawn about sponge placement approaches are not fully supported by the results

The study limitations are not acknowledged

Suggest modifying the Discussion to:

Base conclusions firmly on the results reported

Acknowledge limitations such as the small sample size and retrospective design

Discuss avenues for further research to validate findings

Palande et al. (2017). Negative pressure wound therapy in postoperative spine infections–A systematic review. Asian Spine Journal. This systematic review evaluated the efficacy of NPWT in treating postoperative spinal infections. Though it did not specifically compare sponge placement methods, it represents a highly relevant citation to contextualize the present study's findings within the broader literature on NPWT for DSSI after spinal surgery. discuss and cite discuss and cite https://doi.org/10.4184/asj.2017.11.4.633

Spinal procedures sometimes cause postoperative headaches through risks like dural tears and CSF leakage. But headaches following spine surgery warrant differentiating secondary causes from primary headache disorders to ensure proper management. A thorough history, exam and diagnostic workup are necessary to distinguish secondary headaches from preexisting conditions to accurately diagnose, treat and prevent complications. discuss and cite DOI:10.1007/s00405-021-06724-6

Landmark technique for subclavian central line placement relies on surface anatomy and has a high complication risk in complex spinal surgery patients. Real-time ultrasound guidance enables visualization of relevant structures to precisely place the catheter and may reduce mechanical complications. Several studies show ultrasound guidance provides higher success rates, faster insertion times and fewer complications compared to landmark technique, making it the preferred approach for central line insertion in spinal procedures., discuss and cite doi:10.1097/CCM.0000000000005819

6. PLOS authors have the option to publish the peer review history of their article (what does this mean?). If published, this will include your full peer review and any attached files.

Reviewer #1: No

Reviewer #2: **Yes: **Salvatore

---

## [Author Response · Author response to Decision Letter 0]

30 Aug 2023

Response to Reviewers

Reviewer #1:

Methods

- "After infection control, I used the NPWT technique to promote wound healing with satisfactory results". Please replace "I" with "we".

Answer: Thank you for your kind suggestion. This has been corrected in the revision.

- Please specify whether the study approval by local ethical committee was waived.

Answer: Thank you for your suggestion. We have added “This study has been approved by the 960th hospital of PLA.” in the Methods.

- Please specify the primary and secondary outcomes and provide more details about the statistical analysis performed.

Answer: Thank you for kind suggestion. This has been revised in the new manuscript.

- Authors should describe the intraoperative monitoring used during anesthesia. In fact, several studies have demonstrated the association between the use of intraoperative hemodynamic monitoring and the outcomes after surgery (doi: 10.3390/jcm11020392 - doi: 10.1002/jso.24828). Please discuss and add these 2 references to the list.

Answer: Thank you for your suggestion. Intraoperative monitoring has been discussed and these 2 references has been added in the revision.

Conclusion

- The sentence "NPWT is a safe and effective therapy for deep incision infection following spine surgery that reduces pain, hospitalization time, and expenditures" is not supported by the results. I suggest to soften it, since more data is needed to confirm the results of this study.

Answer: Thank you for your suggestion. This sentence has been revised.

Reviewer #2: perform all the revisions to improve the manuscript: Abstract:

The purpose statement is broad and lacks focus on the sponge implantation strategies being compared

Answer: Thank you for your suggestion. This has been revised, we emphasized the different sponge implantation strategies.

The methods description lacks specificity in study design details

Answer: Thank you for your suggestion. As antibiotic treatment is the fundamental treatment of SSI, we described the antibiotic treatment in the Methods. Although NPWT is an effective method to control the infection, NPWT is not used as a standard procedure to treat SSI, so we described the surgical procedure in the methods. These two parts might cover a part of Methods. We described the different sponge placement in surgical technique, we think this might let the reader know this technique step by step. We also emphasize the sponge placement in the methods. Thank you for your valuable opinion.

The results are reported at a high level without highlighting key findings

Answer: Thank you for your kind suggestion. We have highlighted the key findings in the results.

The conclusion does not state clearly which sponge placement approach shows better outcomes

Answer: Thank you for your valuable opinion. The conclusion in previous manuscript did not state clearly the better outcome of segmental reduction sponge placement. We have revised in this manuscript.

Suggest revising the abstract to:

Precisely state the research question

Detail the methods in an explanatory manner

Concisely highlight key results

Draw a clear conclusion consistent with findings

Note limitations

Answer: Thank you for your opinion. These have been revised in this manuscript.

Introduction:

The background information on NPWT and DSSI is generic rather than focused on the research question.The rationale for comparing centripetal versus segmental sponge placement is not clearly outlined.Suggest providing more targeted background information focused on:

The application of NPWT for DSSI after spinal surgery

The possible differences between centripetal and segmental sponge placement approaches. 

Answer: Thank you for pointing this out. These have been revised in the new manuscript.

Nafamostat mesilate is a synthetic protease inhibitor that can provide anticoagulation during extracorporeal membrane oxygenation (ECMO). Anticoagulation is needed for ECMO circuits to prevent clotting but poses bleeding risks, including for high-risk patients like those undergoing spinal surgery. Nafamostat may offer better control of anticoagulation intensity through its mechanism of inhibiting coagulation factors XIa, XIIa, kallikrein and thrombin. However, research directly examining nafamostat's utility for high bleeding risk patients on ECMO during or after spinal surgery is lacking. More studies are needed to determine if nafamostat can safely enable ECMO support in such complex cases, ideally compared to conventional anticoagulant regimens. The balance of adequate anticoagulation and minimization of bleeding complications will be critical for optimizing ECMO and outcomes in spinal surgery patients., discuss and cite doi: 10.1111/aor.14276

Answer: Thank you for your opinion. I have read this article --Sanfilippo F, Currò JM, La Via L, et al. Use of nafamostat mesilate for anticoagulation during extracorporeal membrane oxygenation: A systematic review. Artif Organs. 2022;46(12):2371-2381. doi:10.1111/aor.14276. Extracorporeal membrane oxygenation (ECMO) represents an advanced option for supporting refractory respiratory and/or cardiac failure. Nafamostat mesilate (NM) is an effective therapy in patients with bleeding risk. The balance of adequate anticoagulation and minimization of bleeding complications is critical in spinal surgery patients. Rivaroxaban was used in patients after spinal surgery. No one was found to have bleeding complications or have complications of deep venous thrombosis. As NPWT did not prevent the patients from routine activities, we advise all patients to do lower extremity functional exercise at bed or walk around the bed.

Wound healing is crucial for patients with deep infections after spinal surgery. NPWT improves outcomes but COVID-19 may interfere. The virus causes excess inflammation and impairs wound repair processes. Though optimizing NPWT through sponge placement may help, COVID-19 poses an added risk factor through viremia, cytokine storming and even COVID-associated vertigo. More studies are needed to evaluate the interaction of COVID-19 with NPWT and wound healing in these vulnerable patients. Finding ways to mitigate COVID's negative effects could maximize NPWT's benefits and minimize complications, shortening hospital stays, lowering costs and improving overall patient outcomes and wellbeing. Further research is warranted to develop effective strategies.discuss and cite doi:10.3389/fmed.2021.790931 

Answer: Thank you for your advice. We agree with you about that COVID-19 may interfere the clinical outcome of NPWT in wound healing. No study has been conducted to investigate the interaction of COVID-19 with NPWT and wound healing in these vulnerable patients. The reason might be the scarce of the patients. The patients who were enrolled in this study was from January 1, 2012 to December 31, 2021. Because of the strict epidemic prevention policy in China, no patient in this group was infected with COVID-19. As the topic of this article (Di Mauro P, La Mantia I, Cocuzza S, et al. Acute Vertigo After COVID-19 Vaccination: Case Series and Literature Review. Front Med (Lausanne). 2022;8:790931. Published 2022 Jan 6. doi:10.3389/fmed.2021.790931) was about the acute vertigo after COVID-19 vaccination, we think it might be better not cited in this article.

Fleischmann et al. (2013). Negative pressure wound therapy after thoracolumbar spinal fusion promotes wound healing and prevents infection: a prospective randomized trial. European Spine Journal. This study compared outcomes between NPWT and standard dressing after spinal fusion surgery. It provides evidence that NPWT reduces wound complications and represents a relevant background article to cite in justifying the use of NPWT for DSSI after spinal surgery. discuss and cite https://doi.org/10.1007/s00586-012-2579-z

Answer: Thank you for your advice. However, this article was not found in the DOI system and in Pubmed. 

Methods:

Important details on study design and sample are lacking (e.g. number of patients in each group, time period of study)

The outcomes being compared between the two groups are not clearly specified

Suggest revising the Methods section to:

Clarify the study design as retrospective

Precisely describe the study sample and size

Explicitly state the outcomes being compared between sponge placement groups

Answer: Thank you for your kind suggestion. These have been revised in the new manuscript.

Results:

Key findings are reported at a high level without numbers and statistics

Figures and tables are referenced but not shown to substantiate claims

Suggest revising the Results section to:

Report numerical data and statistics for all outcome comparisons

Concisely summarize the key findings and outcome differences between groups

Include relevant figures and tables to support claims

Answer: Thank you for your kind suggestion. These have been revised in the new manuscript.

Discussion:

The conclusions drawn about sponge placement approaches are not fully supported by the results

The study limitations are not acknowledged

Suggest modifying the Discussion to:

Base conclusions firmly on the results reported

Acknowledge limitations such as the small sample size and retrospective design

Discuss avenues for further research to validate findings

Answer: Thank you for your kind suggestion. These have been revised in the new manuscript.

Palande et al. (2017). Negative pressure wound therapy in postoperative spine infections–A systematic review. Asian Spine Journal. This systematic review evaluated the efficacy of NPWT in treating postoperative spinal infections. Though it did not specifically compare sponge placement methods, it represents a highly relevant citation to contextualize the present study's findings within the broader literature on NPWT for DSSI after spinal surgery. discuss and cite discuss and cite https://doi.org/10.4184/asj.2017.11.4.633

Answer: Thank you for your advice. However, this article was not found in the DOI system and in Pubmed.

Spinal procedures sometimes cause postoperative headaches through risks like dural tears and CSF leakage. But headaches following spine surgery warrant differentiating secondary causes from primary headache disorders to ensure proper management. A thorough history, exam and diagnostic workup are necessary to distinguish secondary headaches from preexisting conditions to accurately diagnose, treat and prevent complications. discuss and cite DOI:10.1007/s00405-021-06724-6

Answer: Thank you for your advice. Patients would suffer from headache if dural tears and CSF leakage existed. As active cerebrospinal fluid leakage is one of contraindications of NPWT, NPWT was not used if there was dural tear or CSF leakage. No patients enrolled in this study had CSF leakage, some patients complained headache because of the fever, we agree with you that a thorough history, exam and diagnostic workup are necessary to distinguish secondary headaches from preexisting conditions to accurately diagnose, treat and prevent complications. This article (Maniaci A, Merlino F, Cocuzza S, et al. Endoscopic surgical treatment for rhinogenic contact point headache: systematic review and meta-analysis [published correction appears in Eur Arch Otorhinolaryngol. 2021 May 7;:]. Eur Arch Otorhinolaryngol. 2021;278(6):1743-1753. doi:10.1007/s00405-021-06724-6) had been cited in this new manuscript.

Landmark technique for subclavian central line placement relies on surface anatomy and has a high complication risk in complex spinal surgery patients. Real-time ultrasound guidance enables visualization of relevant structures to precisely place the catheter and may reduce mechanical complications. Several studies show ultrasound guidance provides higher success rates, faster insertion times and fewer complications compared to landmark technique, making it the preferred approach for central line insertion in spinal procedures., discuss and cite doi:10.1097/CCM.0000000000005819

Answer: Thank you for you opinion. As the topic was NPWT in patients with DSSI, the use of ultrasound guidance to place catheter might be irrelevant, so we think it might be inappropriate to discuss and cite this article. Thank you for your suggestion.

---

## [Editor Report · Decision Letter 1]

7 Sep 2023

Effects of Different Sponge Implantation Methods of Negative Pressure Wound Therapy on Wound Healing of Deep Surgical Site Infection After Spinal Surgery

PONE-D-23-14685R1

Dear Dr. Chang,

We’re pleased to inform you that your manuscript has been judged scientifically suitable for publication and will be formally accepted for publication once it meets all outstanding technical requirements.

Kind regards,

Antonino Maniaci

Academic Editor

PLOS ONE

Additional Editor Comments (optional):

Well done, all the suggestions required were addressed and the paper is improved. Bests
---

## [Editor Report · Acceptance letter]

20 Sep 2023

PONE-D-23-14685R1 

Effects of Different Sponge Implantation Methods of Negative Pressure Wound Therapy on Wound Healing of Deep Surgical Site Infection After Spinal Surgery 

Dear Dr. Chang:

I'm pleased to inform you that your manuscript has been deemed suitable for publication in PLOS ONE. Congratulations! Your manuscript is now with our production department. 

Kind regards, 

on behalf of

Dr. Antonino Maniaci 

Academic Editor

PLOS ONE